# Learning to Draw Samples: With Application to Amortized MLE for Generative Adversarial Learning

**Dilin Wang,   Qiang Liu**
Department of Computer Science, Dartmouth College
{dilin.wang.gr, qiang.liu}@dartmouth.edu

## Abstract

We propose a simple algorithm to train stochastic neural networks to draw samples from given target distributions for probabilistic inference. Our method is based on iteratively adjusting the neural network parameters so that the output changes along a Stein variational gradient (Liu & Wang, 2016) that maximumly decreases the KL divergence with the target distribution. Our method works for any target distribution specified by their unnormalized density function, and can train any black-box architectures that are differentiable in terms of the parameters we want to adapt. As an application of our method, we propose an *amortized MLE* algorithm for training deep energy model, where a neural sampler is adaptively trained to approximate the likelihood function. Our method mimics an adversarial game between the deep energy model and the neural sampler, and obtains realistic-looking images competitive with the state-of-the-art results.

## 1   Introduction

Modern machine learning increasingly relies on highly complex probabilistic models to reason about uncertainty. A key computational challenge is to develop efficient inference techniques to approximate, or draw samples from complex distributions. Currently, most inference methods, including MCMC and variational inference, are hand-designed by researchers or domain experts. This makes it difficult to fully optimize the choice of different methods and their parameters, and exploit the structures in the problems of interest in an automatic way. The hand-designed algorithm can also be inefficient when it requires to make fast inference repeatedly on a large number of different distributions with similar structures. This happens, for example, when we need to reason about a number of observed datasets in settings like online learning, or need fast inference as inner loops for other algorithms such as maximum likelihood training. Therefore, it is highly desirable to develop more intelligent probabilistic inference systems that can adaptively improve its own performance to fully the optimize computational efficiency, and generalize to new tasks with similar structures.

Specifically, denote by $p(x)$ a probability density of interest specified up to the normalization constant, which we want to draw sample from, or marginalize to estimate its normalization constant. We want to study the following problem:

**Problem 1.** *Given a distribution with density $p(x)$ and a function $f(\eta; \xi)$ with parameter $\eta$ and random input $\xi$, for which we only have assess to draws of the random input $\xi$ (without knowing its true distribution $q_0$), and the output values of $f(\eta; \xi)$ and its derivative $\partial_\eta f(\eta; \xi)$ given $\eta$ and $\xi$. We want to find an optimal parameter $\eta$ so that the density of the random output variable $x = f(\eta; \xi)$ with $\xi \sim q_0$ closely matches the target density $p(x)$.*

Because we have no assumption on the structure of $f(\eta; \xi)$ and the distribution of random input, we can not directly calculate the actual distribution of the output random variable $x = f(\eta; \xi)$; this makes it difficult to solve Problem 1 using the traditional variational inference (VI) methods. Recall that traditional VI approximates $p(x)$ using simple proposal distributions $q_\eta(x)$ indexed by parameter $\eta$, and finds the optimal $\eta$ by minimizing KL divergence $\mathrm{KL}(q_\eta \,||\, p) = \mathbb{E}_{q_\eta}[\log(q_\eta/p)]$, which requires to calculate the density $q_\eta(x)$ or its derivative that is not computable by our assump-

tion (even when the Monte Carlo gradient estimation and the reparametrization trick (Kingma & Welling, 2013) are applied).

In fact, it is this requirement of calculating $q_\eta(x)$ that has been the major constraint for the designing of state-of-the-art variational inference methods with rich approximation families; the recent successful algorithms (e.g., Rezende & Mohamed, 2015b; Tran et al., 2015; Ranganath et al., 2015, to name only a few) have to handcraft special variational families to ensure the computational tractability of $q_\eta(x)$ and simultaneously obtain high approximation accuracy, which require substantial mathematical insights and research effects. Methods that do not require to explicitly calculate $q_\eta(x)$ can significantly simplify the design and applications of VI methods, allowing practical users to focus more on choosing proposals that work best with their specific tasks. We will use the term *wild variational inference* to refer to new variants of variational methods that require no tractability $q_\eta(x)$, to distinguish with the *black-box variational inference* (Ranganath et al., 2014) which refers to methods that work for generic target distributions $p(x)$ without significant model-by-model consideration (but still require to calculate the proposal density $q_\eta(x)$).

A similar problem also appears in importance sampling (IS), where it requires to calculate the IS proposal density $q(x)$ in order to calculate the importance weight $w(x) = p(x)/q(x)$. However, there exist methods that use no explicit information of $q(x)$, which, seemingly counter-intuitively, give better asymptotic variance or converge rates than the typical IS that uses the proposal information (e.g., Liu & Lee, 2016; Briol et al., 2015; Henmi et al., 2007; Delyon & Portier, 2014). Discussions on this phenomenon dates back to O'Hagan (1987), who argued that "Monte Carlo (that uses the proposal information) is fundamentally unsound" for violating the Likelihood Principle, and developed Bayesian Monte Carlo (O'Hagan, 1991) as an example that uses no information on $q(x)$, yet gives better convergence rate than the typical Monte Carlo $O(n^{-1/2})$ rate (Briol et al., 2015). Despite the substantial difference between IS and VI, these results intuitively suggest the possibility of developing efficient variational inference without calculating $q(x)$ explicitly.

In this work, we propose a simple algorithm for Problem 1 by iteratively adjusting the network parameter $\eta$ to make its output random variable changes along a Stein variational gradient direction (SVGD) (Liu & Wang, 2016) that optimally decreases its KL divergence with the target distribution. Critically, the SVGD gradient includes a repulsive term to ensure that the generated samples have the right amount of variability that matches $p(x)$. In this way, we "amortize SVGD" using a neural network, which makes it possible for our method to adaptively improve its own efficiency by leveraging fast experience, especially in cases when it needs to perform fast inference repeatedly on a large number of similar tasks. As an application, we use our method to amortize the MLE training of deep energy models, where a neural sampler is adaptively trained to approximate the likelihood function. Our method, which we call *SteinGAN*, mimics an adversarial game between the energy model and the neural sampler, and obtains realistic-looking images competitive with the state-of-the-art results produced by generative adversarial networks (GAN) (Goodfellow et al., 2014; Radford et al., 2015).

**Related Work**  The idea of amortized inference (Gershman & Goodman, 2014) has been recently applied in various domains of probabilistic reasoning, including both amortized variational inference (e.g., Kingma & Welling, 2013; Rezende & Mohamed, 2015a), and data-driven proposals for (sequential) Monte Carlo methods (e.g., Paige & Wood, 2016), to name only a few. Most of these methods, however, require to explicitly calculate $q(x)$ (or its gradient). One exception is a very recent paper (Ranganath et al., 2016) that avoids calculating $q(x)$ using an idea related to Stein discrepancy (Gorham & Mackey, 2015; Liu et al., 2016; Oates et al., 2014; Chwialkowski et al., 2016). There is also a raising interest recently on a similar problem of "learning to optimize" (e.g., Andrychowicz et al., 2016; Daniel et al., 2016; Li & Malik, 2016), which is technically easier than the more general problem of "learning to sample". In fact, we show that our algorithm reduces to "learning to optimize" when only one particle is used in SVGD.

Generative adversarial network (GAN) and its variants have recently gained remarkable success on generating realistic-looking images (Goodfellow et al., 2014; Salimans et al., 2016; Radford et al., 2015; Li et al., 2015; Dziugaite et al., 2015; Nowozin et al., 2016). All these methods are set up to train latent variable models (the generator) under the assistant of the discriminator. Our SteinGAN instead performs traditional MLE training for a deep energy model, with the help of a neural sampler that learns to draw samples from the energy model to approximate the likelihood

function; this admits an adversarial interpretation: we can view the neural sampler as a generator that attends to fool the deep energy model, which in turn serves as a discriminator that distinguishes the real samples and the simulated samples given by the neural sampler. This idea of training MLE with neural samplers was first discussed by Kim & Bengio (2016); one of the key differences is that the neural sampler in Kim & Bengio (2016) is trained with the help of a heuristic diversity regularizer based on batch normalization, while SVGD enforces the diversity in a more principled way. Another method by Zhao et al. (2016) also trains an energy score to distinguish real and simulated samples, but within a non-probabilistic framework (see Section 5 for more discussion). Other more traditional approaches for training energy-based models (e.g., Ngiam et al., 2011; Xie et al., 2016) are often based on variants of MCMC-MLE or contrastive divergence (Geyer, 1991; Hinton, 2002; Tieleman, 2008), and have difficulty generating realistic-looking images from scratch.

## 2 STEIN VARIATIONAL GRADIENT DESCENT (SVGD)

Stein variational gradient descent (SVGD) (Liu & Wang, 2016) is a general purpose Bayesian inference algorithm motivated by Stein's method (Stein, 1972; Barbour & Chen, 2005) and kernelized Stein discrepancy (Liu et al., 2016; Chwialkowski et al., 2016; Oates et al., 2014). It uses an efficient *deterministic* gradient-based update to iteratively evolve a set of particles $\{x_i\}_{i=1}^n$ to minimize the KL divergence with the target distribution. SVGD has a simple form that reduces to the typical gradient descent for maximizing $\log p$ when using only one particle ($n = 1$), and hence can be easily combined with the successful tricks for gradient optimization, including stochastic gradient, adaptive learning rates (such as adagrad), and momentum.

To give a quick overview of the main idea of SVGD, let $p(x)$ be a positive density function on $\mathbb{R}^d$ which we want to approximate with a set of particles $\{x_i\}_{i=1}^n$. SVGD initializes the particles by sampling from some simple distribution $q_0$, and updates the particles iteratively by

$$x_i \leftarrow x_i + \epsilon \phi(x_i), \quad \forall i = 1, \ldots, n, \tag{1}$$

where $\epsilon$ is a step size, and $\phi(x)$ is a "particle gradient direction" chosen to maximumly decrease the KL divergence between the distribution of particles and the target distribution, in the sense that

$$\phi = \underset{\phi \in \mathcal{F}}{\arg \max} \left\{ -\frac{d}{d\epsilon} \mathrm{KL}(q_{[\epsilon\phi]} \,\|\, p)\big|_{\epsilon=0} \right\}, \tag{2}$$

where $q_{[\epsilon\phi]}$ denotes the density of the updated particle $x' = x + \epsilon\phi(x)$ when the density of the original particle $x$ is $q$, and $\mathcal{F}$ is the set of perturbation directions that we optimize over. We choose $\mathcal{F}$ to be the unit ball of a vector-valued reproducing kernel Hilbert space (RKHS) $\mathcal{H}^d = \mathcal{H} \times \cdots \times \mathcal{H}$ with each $\mathcal{H}$ associating with a positive definite kernel $k(x, x')$; note that $\mathcal{H}$ is dense in the space of continuous functions with universal kernels such as the Gaussian RBF kernel.

Critically, the gradient of KL divergence in (2) equals a simple linear functional of $\phi$, allowing us to obtain a closed form solution for the optimal $\phi$. Liu & Wang (2016) showed that

$$-\frac{d}{d\epsilon} \mathrm{KL}(q_{[\epsilon\phi]} \,\|\, p)\big|_{\epsilon=0} = \mathbb{E}_{x \sim q}[\mathcal{T}_p \phi(x)], \tag{3}$$

$$\text{with} \quad \mathcal{T}_p \phi(x) = \nabla_x \log p(x)^\top \phi(x) + \nabla_x \cdot \phi(x), \tag{4}$$

where $\mathcal{T}_p$ is considered as a linear operator acting on function $\phi$ and is called the Stein operator in connection with Stein's identity which shows that the RHS of (3) equals zero if $p = q$:

$$\mathbb{E}_p[\mathcal{T}_p \phi] = \mathbb{E}_p[\nabla_x \log p^\top \phi + \nabla_x \cdot \phi] = 0. \tag{5}$$

This is a result of integration by parts assuming the value of $p(x)\phi(x)$ vanishes on the boundary of the integration domain.

Therefore, the optimization in (2) reduces to

$$\mathbb{D}(q \,\|\, p) \overset{def}{=} \underset{\phi \in \mathcal{H}^d}{\max} \{ \mathbb{E}_{x \sim q}[\mathcal{T}_p \phi(x)] \quad s.t. \quad \|\phi\|_{\mathcal{H}^d} \le 1 \}, \tag{6}$$

where $\mathbb{D}(q \,\|\, p)$ is the kernelized Stein discrepancy defined in Liu et al. (2016), which equals zero if and only if $p = q$ under mild regularity conditions. Importantly, the optimal solution of (6) yields a closed form

$$\phi^*(x') \propto \mathbb{E}_{x \sim q}[\nabla_x \log p(x) k(x, x') + \nabla_x k(x, x')].$$

---

**Algorithm 1** Amortized SVGD for Problem 1

---
Set batch size $m$, step-size scheme $\{\epsilon_t\}$ and kernel $k(x, x')$. Initialize $\eta^0$.
**for** iteration $t$ **do**
 Draw random $\{\xi_i\}_{i=1}^m$, calculate $x_i = f(\eta^t; \xi_i)$, and the Stein variational gradient $\Delta x_i$ in (7).
 Update parameter $\eta$ using (8), (9) or (10).
**end for**

---

By approximating the expectation under $q$ with the empirical average of the current particles $\{x_i\}_{i=1}^n$, SVGD admits a simple form of update:

$$x_i \leftarrow x_i + \epsilon \Delta x_i, \qquad \forall i = 1, \ldots, n,$$

$$\text{where} \quad \Delta x_i = \hat{\mathbb{E}}_{x \in \{x_i\}_{i=1}^n}[\nabla_x \log p(x) k(x, x_i) + \nabla_x k(x, x_i)], \tag{7}$$

and $\hat{\mathbb{E}}_{x \sim \{x_i\}_{i=1}^n}[f(x)] = \sum_i f(x_i)/n$. The two terms in $\Delta x_i$ play two different roles: the term with the gradient $\nabla_x \log p(x)$ drives the particles toward the high probability regions of $p(x)$, while the term with $\nabla_x k(x, x_i)$ serves as a repulsive force to encourage diversity; to see this, consider a stationary kernel $k(x, x') = k(x - x')$, then the second term reduces to $\hat{\mathbb{E}}_x \nabla_x k(x, x_i) = -\hat{\mathbb{E}}_x \nabla_{x_i} k(x, x_i)$, which can be treated as the negative gradient for minimizing the average similarity $\hat{\mathbb{E}}_x k(x, x_i)$ in terms of $x_i$. Overall, this particle update produces diverse points for distributional approximation and uncertainty assessment, and also has an interesting "momentum" effect in which the particles move collaboratively to escape the local optima.

It is easy to see from (7) that $\Delta x_i$ reduces to the typical gradient $\nabla_x \log p(x_i)$ when there is only a single particle ($n = 1$) and $\nabla_x k(x, x_i)$ when $x = x_i$, in which case SVGD reduces to the standard gradient ascent for maximizing $\log p(x)$ (i.e., maximum *a posteriori* (MAP)).

## 3 Amortized SVGD: Towards an Automatic Neural Sampler

SVGD and other particle-based methods become inefficient when we need to repeatedly infer a large number different target distributions for multiple tasks, including online learning or inner loops of other algorithms, because they can not improve based on the experience from the past tasks, and may require a large memory to restore a large number of particles. We propose to "amortize SVGD" by training a neural network $f(\eta; \xi)$ to mimic the SVGD dynamics, yielding a solution for Problem 1.

One straightforward way to achieve this is to run SVGD to convergence and train $f(\eta; \xi)$ to fit the SVGD results. This, however, requires to run many epochs of fully converged SVGD and can be slow in practice. We instead propose an *incremental approach* in which $\eta$ is iteratively adjusted so that the network outputs $x = f(\eta; \xi)$ changes along the Stein variational gradient direction in (7) in order to decrease the KL divergence between the target and approximation distribution.

To be specific, denote by $\eta^t$ the estimated parameter at the $t$-th iteration of our method; each iteration of our method draws a batch of random inputs $\{\xi_i\}_{i=1}^m$ and calculate their corresponding output $x_i = f(\eta; \xi_i)$ based on $\eta^t$; here $m$ is a mini-batch size (e.g., $m = 100$). The Stein variational gradient $\Delta x_i$ in (7) would then ensure that $x_i' = x_i + \epsilon \Delta x_i$ forms a better approximation of the target distribution $p$. Therefore, we should adjust $\eta$ to make its output matches $\{x_i'\}$, that is, we want to update $\eta$ by

$$\eta^{t+1} \leftarrow \arg\min_\eta \sum_{i=1}^m ||f(\eta; \xi_i) - x_i'||_2^2, \quad \text{where} \quad x_i' = x_i + \epsilon \Delta x_i. \tag{8}$$

See Algorithm 1 for the summary of this procedure. If we assume $\epsilon$ is very small, then (8) reduces to a least square optimization. To see this, note that $f(\eta; \xi_i) \approx f(\eta^t; \xi_i) + \partial_\eta f(\eta^t; \xi_i)(\eta - \eta^t)$ by Taylor expansion. Since $x_i = f(\eta^t; \xi_i)$, we have

$$||f(\eta; \xi_i) - x_i'||_2^2 \approx ||\partial_\eta f(\eta^t; \xi_i)(\eta - \eta^t) - \epsilon \Delta x_i||_2^2.$$

As a result, (8) reduces to the following least square optimization:

$$\eta^{t+1} \leftarrow \eta^t + \epsilon \Delta \eta^t, \quad \text{where} \quad \Delta \eta^t = \arg\min_\delta \sum_{i=1}^m ||\partial_\eta f(\eta^t; \xi_i)\delta - \Delta x_i||_2^2. \tag{9}$$

Update (9) can still be computationally expensive because of the matrix inversion. We can derive a further approximation by performing only one step of gradient descent of (8) (or (9)), which gives

$$\eta^{t+1} \leftarrow \eta^t + \epsilon \sum_{i=1}^{m} \partial_\eta f(\eta^t; \xi_i)\Delta x_i. \tag{10}$$

Although update (10) is derived as an approximation of (8)-(9), it is computationally faster and we find it works very effectively in practice; this is because when $\epsilon$ is small, one step of gradient update can be sufficiently close to the optimum.

Update (10) also has a simple and intuitive form: (10) can be thought as *a "chain rule" that back-propagates the Stein variational gradient to the network parameter $\eta$*. This can be justified by considering the special case when we use only a single particle ($n = 1$) in which case $\Delta x_i$ in (7) reduces to the typical gradient $\nabla_x \log p(x_i)$ of $\log p(x)$, and update (10) reduces to the typical gradient ascent for maximizing

$$\mathbb{E}_\xi[\log p(f(\eta; \xi))],$$

in which case $f(\eta; \xi)$ is trained to maximize $\log p(x)$ (that is, *learning to optimize*), instead of *learning to draw samples from $p$* for which it is crucial to use Stein variational gradient $\Delta x_i$ to diversify the network outputs.

Update (10) also has a close connection with the typical variational inference with the reparameterization trick (Kingma & Welling, 2013). Let $q_\eta(x)$ be the density function of $x = f(\eta; \xi), \xi \sim q_0$. Using the reparameterization trick, the gradient of $\text{KL}(q_\eta \| p)$ w.r.t. $\eta$ can be shown to be

$$\nabla_\eta \text{KL}(q_\eta \| p) = -\mathbb{E}_{\xi \sim q_0}[\partial_\eta f(\eta; \xi)(\nabla_x \log p(x) - \nabla_x \log q_\eta(x))].$$

With $\{\xi_i\}$ i.i.d. drawn from $q_0$ and $x_i = f(\eta; \xi_i)$, $\forall i$, the standard stochastic gradient descent for minimizing the KL divergence is

$$\eta^{t+1} \leftarrow \eta^t + \sum_i \partial_\eta f(\eta^t; \xi_i)\tilde{\Delta} x_i, \quad \text{where} \quad \tilde{\Delta} x_i = \nabla_x \log p(x_i) - \nabla_x \log q_\eta(x_i). \tag{11}$$

This is similar with (10), but replaces the Stein gradient $\Delta x_i$ defined in (7) with $\tilde{\Delta} x_i$. The advantage of using $\Delta x_i$ is that it does not require to explicitly calculate $q_\eta$, and hence admits a solution to Problem 1 in which $q_\eta$ is not computable for complex network $f(\eta; \xi)$ and unknown input distribution $q_0$. Further insights can be obtained by noting that

$$\begin{aligned} \Delta x_i &\approx \mathbb{E}_{x \sim q}[\nabla_x \log p(x)k(x, x_i) + \nabla_x k(x, x_i)] \\ &= \mathbb{E}_{x \sim q}[(\nabla_x \log p(x) - \nabla_x \log q(x))k(x, x_i)] \\ &= \mathbb{E}_{x \sim q}[(\tilde{\Delta} x)k(x, x_i)], \end{aligned} \tag{12}$$

where (12) is obtained by using Stein's identity (5). Therefore, $\Delta x_i$ can be treated as a kernel smoothed version of $\tilde{\Delta} x_i$.

## 4  AMORTIZED MLE FOR GENERATIVE ADVERSARIAL TRAINING

Our method allows us to design efficient approximate sampling methods adaptively and automatically, and enables a host of novel applications. In this paper, we apply it in an amortized MLE method for training deep generative models.

Maximum likelihood estimator (MLE) provides a fundamental approach for learning probabilistic models from data, but can be computationally prohibitive on distributions for which drawing samples or computing likelihood is intractable due to the normalization constant. Traditional methods such as MCMC-MLE use hand-designed methods (e.g., MCMC) to approximate the intractable likelihood function but do not work efficiently in practice. We propose to adaptively train a generative neural network to draw samples from the distribution during MLE training, which not only provides computational advantage, and also allows us to generate realistic-looking images competitive with, or better than the state-of-the-art generative adversarial networks (GAN) (Goodfellow et al., 2014; Radford et al., 2015) (see Figure 1-5).

---

**Algorithm 2** Amortized MLE as Generative Adversarial Learning

---

**Goal:** MLE training for energy model $p(x|\theta) = \exp(-\phi(x, \theta) - \Phi(\theta))$.

Initialize $\eta$ and $\theta$.

**for** iteration $t$ **do**

 **Updating $\eta$:** Draw $\xi_i \sim q_0$, $x_i = f(\eta; \xi_i)$; update $\eta$ using (8), (9) or (10) with $p(x) = p(x|\theta)$. Repeat several times when needed.

 **Updating $\theta$:** Draw a mini-batch of observed data $\{x_{i,obs}\}$, and simulated data $x_i = f(\eta; \xi_i)$, update $\theta$ by (13).

**end for**

---

To be specific, denote by $\{x_{i,obs}\}$ a set of observed data. We consider the maximum likelihood training of energy-based models of form

$$p(x|\theta) = \exp(-\phi(x, \theta) - \Phi(\theta)), \quad \Phi(\theta) = \log \int \exp(-\phi(x, \theta))dx,$$

where $\phi(x; \theta)$ is an energy function for $x$ indexed by parameter $\theta$ and $\Phi(\theta)$ is the log-normalization constant. The log-likelihood function of $\theta$ is

$$L(\theta) = \frac{1}{n}\sum_{i=1}^{n} \log p(x_{i,obs}|\theta),$$

whose gradient is

$$\nabla_\theta L(\theta) = -\hat{\mathbb{E}}_{obs}[\partial_\theta \phi(x; \theta)] + \mathbb{E}_\theta[\partial_\theta \phi(x; \theta)],$$

where $\hat{\mathbb{E}}_{obs}[\cdot]$ and $\mathbb{E}_\theta[\cdot]$ denote the empirical average on the observed data $\{x_{i,obs}\}$ and the expectation under model $p(x|\theta)$, respectively. The key computational difficulty is to approximate the model expectation $\mathbb{E}_\theta[\cdot]$. To address this problem, we use a generative neural network $x = f(\eta; \xi)$ trained by Algorithm 1 to approximately sample from $p(x|\theta)$, yielding a gradient update for $\theta$ of form

$$\theta \leftarrow \theta + \epsilon\hat{\nabla}_\theta L(\theta), \qquad \hat{\nabla}_\theta L(\theta) = -\hat{\mathbb{E}}_{obs}[\partial_\theta \phi(x; \theta)] + \hat{\mathbb{E}}_\eta[\partial_\theta \phi(x; \theta)], \tag{13}$$

where $\hat{\mathbb{E}}_\eta$ denotes the empirical average on $\{x_i\}$ where $x_i = f(\eta; \xi_i)$, $\{\xi_i\} \sim q_0$. As $\theta$ is updated by gradient ascent, $\eta$ is successively updated via Algorithm 1 to *follow $p(x|\theta)$*. See Algorithm 2.

We call our method *SteinGAN*, because it can be intuitively interpreted as an adversarial game between the generative network $f(\eta; \xi)$ and the energy model $p(x|\theta)$ which serves as a discriminator: The MLE gradient update of $p(x|\theta)$ effectively decreases the energy of the training data and increases the energy of the simulated data from $f(\eta; \xi)$, while the SVGD update of $f(\eta; \xi)$ decreases the energy of the simulated data to fit better with $p(x|\theta)$. Compared with the traditional methods based on MCMC-MLE or contrastive divergence, we *amortize the sampler as we train*, which gives much faster speed and simultaneously provides a high quality generative neural network that can generate realistic-looking images; see Kim & Bengio (2016) for a similar idea and discussions.

## 5 EMPIRICAL RESULTS

We evaluated our SteinGAN on four datasets, MNIST, CIFAR-10, CelebA (Liu et al., 2015), and Large-scale Scene Understanding (LSUN) (Yu et al., 2015), on which we find our method tends to generate realistic-looking images competitive with, sometimes better than DCGAN (Radford et al., 2015) (see Figure 2 - Figure 3). Our code is available at `https://github.com/DartML/SteinGAN`.

**Model Setup** In order to generate realistic-looking images, we define our energy model based on an autoencoder:

$$p(x|\theta) \propto \exp(-||x - \mathrm{D}(\mathrm{E}(x; \theta); \theta)||), \tag{14}$$

where $x$ denotes the image. This choice is motivated by Energy-based GAN (Zhao et al., 2016) in which the autoencoder loss is used as a discriminator but without a probabilistic interpretation. We

assume $f(\eta; \xi)$ to be a neural network whose input $\xi$ is a 100-dimensional random vector drawn by Uniform$([-1, 1])$. The positive definite kernel in SVGD is defined by the RBF kernel on the hidden representation obtained by the autoencoder in (14), that is,

$$k(x, x') = \exp(-\frac{1}{h^2}||\mathrm{E}(x; \theta) - \mathrm{E}(x'; \theta)||^2).$$

As it is discussed in Section 3, the kernel provides a repulsive force to produce an amount of variability required for generating samples from $p(x)$. This is similar to the heuristic repelling regularizer in Zhao et al. (2016) and the batch normalization based regularizer in Kim & Bengio (2016), but is derived in a more principled way. We take the bandwidth to be $h = 0.5 \times \mathrm{med}$, where $\mathrm{med}$ is the median of the pairwise distances between $\mathrm{E}(x)$ on the image simulated by $f(\eta; \xi)$. This makes the kernel change adaptively based on both $\theta$ (through $\mathbb{E}(x; \theta)$) and $\eta$ (through bandwidth $h$).

Some datasets include both images $x$ and their associated discrete labels $y$. In these cases, we train a joint energy model on $(x, y)$ to capture both the inner structure of the images and its predictive relation with the label, allowing us to simulate images with a control on which category it belongs to. Our joint energy model is defined to be

$$p(x, y|\theta) \propto \exp\left\{ -||x - \mathrm{D}(\mathrm{E}(x; \theta); \theta)|| - \max[m, \sigma(y, \mathrm{E}(x; \theta))] \right\}, \qquad (15)$$

where $\sigma(\cdot, \cdot)$ is the cross entropy loss function of a fully connected output layer. In this case, our neural sampler first draws a label $y$ randomly according to the empirical counts in the dataset, and then passes $y$ into a neural network together with a $100 \times 1$ random vector $\xi$ to generate image $x$. This allows us to generate images for particular categories by controlling the value of input $y$.

**Stabilization** In practice, we find it is useful to modify (13) to be

$$\theta \leftarrow \theta - \epsilon\hat{\mathbb{E}}_{obs}[\nabla_\theta \phi(x, \theta)] + \epsilon(1 - \gamma)\hat{\mathbb{E}}_\eta[\nabla_\theta \phi(x, \theta)]. \qquad (16)$$

where $\gamma$ is a discount factor (which we take to be $\gamma = 0.7$). This is equivalent to maximizing a regularized likelihood:

$$\max_\theta \{\log p(x|\theta) + \gamma\Phi(\theta)\}$$

where $\Phi(\theta)$ is the log-partition function; note that $\exp(\gamma\Phi(\theta))$ is a conjugate prior of $p(x|\theta)$.

We initialize the weights of both the generator and discriminator from Gaussian distribution $\mathcal{N}(0, 0.02)$, and train them using Adam (Kingma & Ba, 2014) with a learning rate of $0.001$ for the generator and $0.0001$ for the energy model (the discriminator). In order to keep the generator and discriminator approximately aligned during training, we speed up the MLE update (16) of the discriminator (by increasing its learning rate to $0.0005$) when the energy of the real data batch is larger than the energy of the simulated images, while slow down it (by freezing the MLE update of $\theta$ in (16)) if the magnitude of the energy difference between the real images and the simulated images goes above a threshold of $0.5$. We used the bag of architecture guidelines for stable training suggested in DCGAN (Radford et al., 2015).

**Discussion** The MNIST dataset has a training set of $60,000$ examples. Both DCGAN and our model produce high quality images, both visually indistinguishable from real images; see figure 1.

CIFAR-10 is very diverse, and with only $50,000$ training examples. Figure 2 shows examples of simulated images by DCGAN and SteinGAN generated conditional on each category, which look equally well visually. We also provide quantitively evaluation using a recently proposed inception score (Salimans et al., 2016), as well as the classification accuracy when training ResNet using $50,000$ simulated images as train sets, evaluated on a separate held-out testing set never seen by the GAN models. Besides DCGAN and SteinGAN, we also evaluate another simple baseline obtained by subsampling 500 real images from the training set and duplicating them 100 times. We observe that these scores capture rather different perspectives of image generation: The inception score favors images that look realistic individually and have uniformly distributed labels; as a result, the inception score of the duplicated 500 images is almost as high as the real training set. We find that the inception score of SteinGAN is comparable, or slightly lower than that of DCGAN. On the other hand, the classification accuracy measures the amount information captured in the simulated image sets; we find that SteinGAN achieves the highest classification accuracy, suggesting that it captures more information in the training set.

Figure 3 and 4 visualize the results on CelebA (with more than 200k face images) and LSUN (with nearly 3M bedroom images), respectively. We cropped and resized both dataset images into $64 \times 64$.

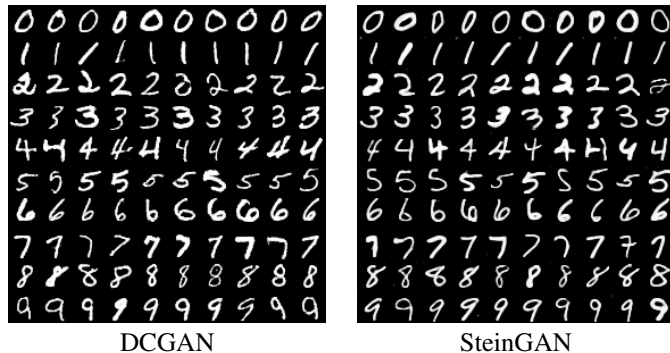

DCGAN                                    SteinGAN

Figure 1: MNIST images generated by DCGAN and our SteinGAN. We use the joint model in (15) to allow us to generate images for each digit. We set $m = 0.2$.

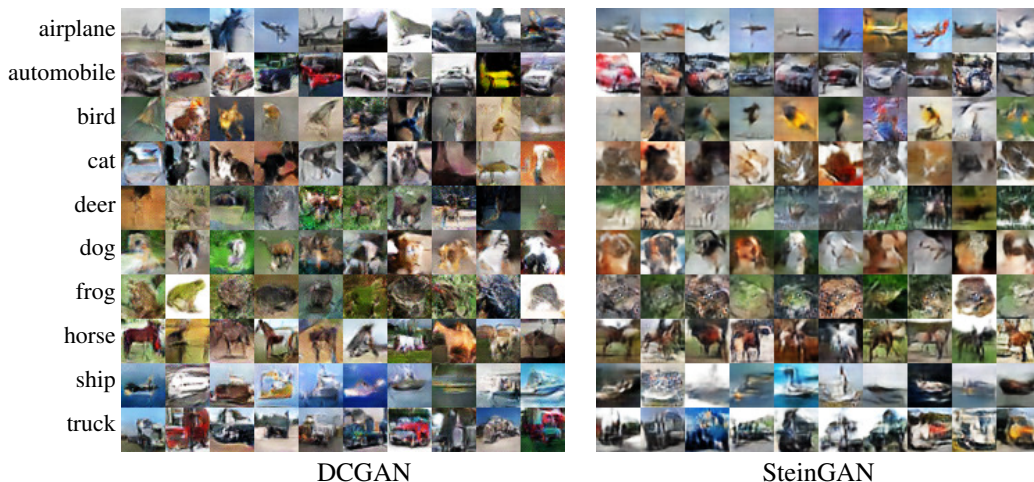

DCGAN                                    SteinGAN

| Inception Score | | | | |
|---|---|---|---|---|
| | Real Training Set | 500 Duplicate | DCGAN | SteinGAN |
| Model Trained on ImageNet | 11.237 | 11.100 | 6.581 | 6.351 |
| Model Trained on CIFAR-10 | 9.848 | 9.807 | 7.368 | 7.428 |

| Testing Accuracy | | | |
|---|---|---|---|
| Real Training Set | 500 Duplicate | DCGAN | SteinGAN |
| 92.58 % | 44.96 % | 44.78 % | 63.81 % |

Figure 2: Results on CIFAR-10. "500 Duplicate" denotes 500 images randomly subsampled from the training set, each duplicated 100 times. Upper: images simulated by DCGAN and SteinGAN (based on joint model (15)) conditional on each category. Middle: inception scores for samples generated by various methods (all with 50,000 images) on inception models trained on ImageNet and CIFAR-10, respectively. Lower: testing accuracy on real testing set when using 50,000 simulated images to train ResNets for classification. SteinGAN achieves higher testing accuracy than DCGAN. We set $m = 1$ and $\gamma = 0.8$.

## 6 CONCLUSION

We propose a new method to train neural samplers for given distributions, together with a new SteinGAN method for generative adversarial training. Future directions involve more applications and theoretical understandings for training neural samplers.

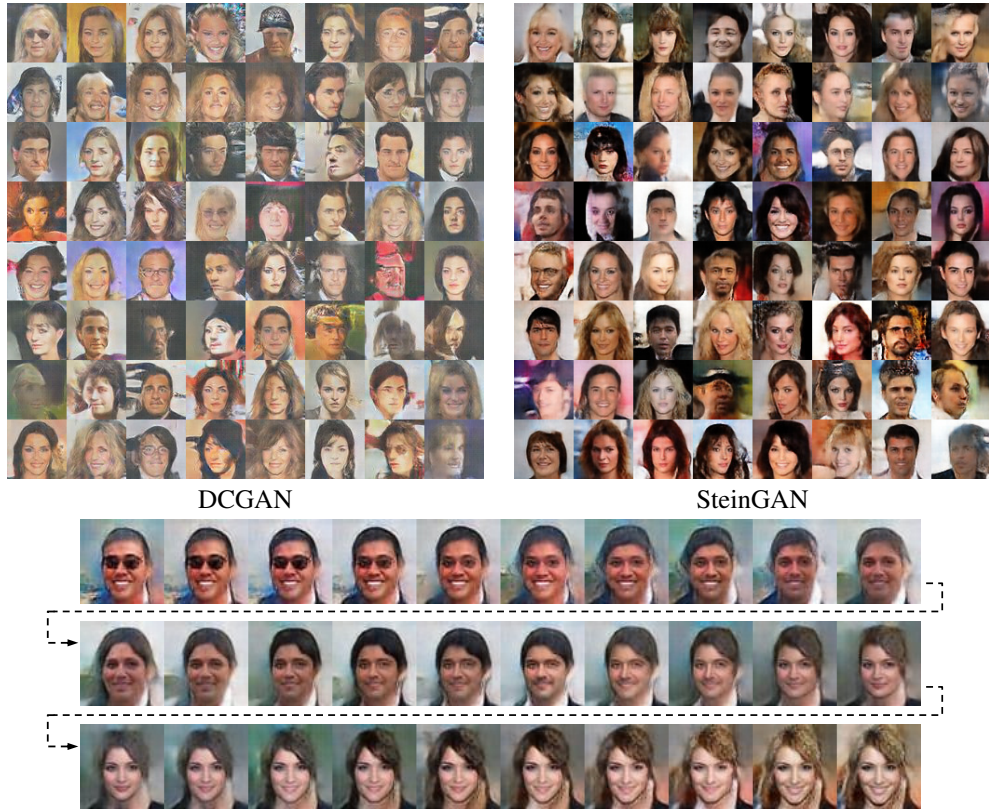

DCGAN                                                    SteinGAN

Figure 3: Results on CelebA. Upper: images generated by DCGAN and our SteinGAN. Lower: images generated by SteinGAN when performing a random walk $\xi \leftarrow \xi + 0.01 \times \mathrm{Uniform}([-1, 1])$ on the random input $\xi$; we can see that a man with glasses and black hair gradually changes to a woman with blonde hair. See Figure 5 for more examples.

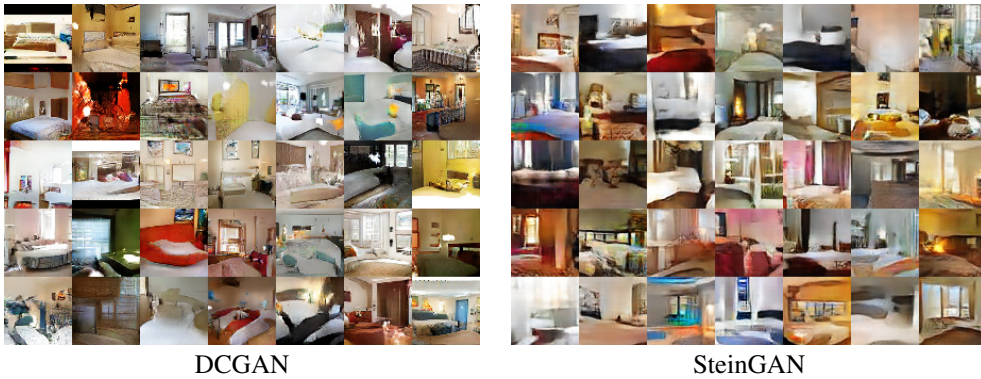

DCGAN                                                    SteinGAN

Figure 4: Images generated by DCGAN and our SteinGAN on LSUN.

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

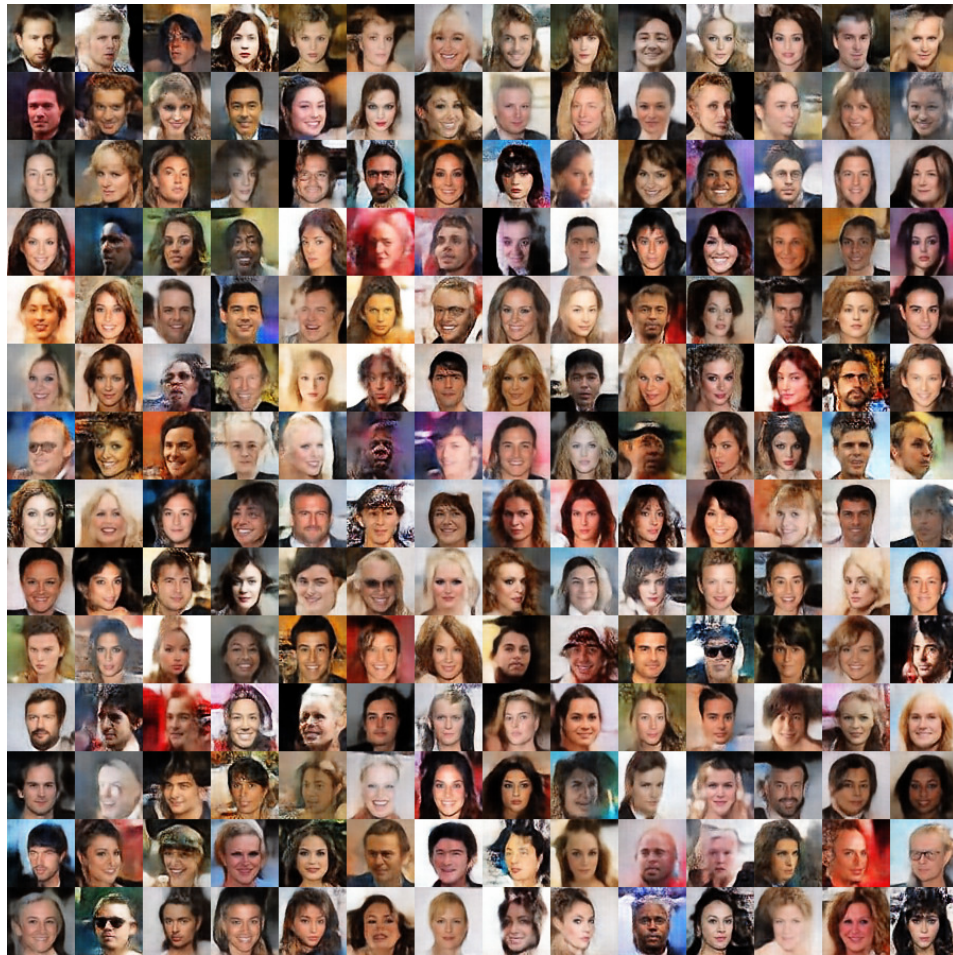

Figure 5: More images generated by SteinGAN on CelebA.

