# Peer review of "Learning to Draw Samples: With Application to Amortized MLE for Generative Adversarial Learning"

_ICLR 2017 — rejected_

[Official Review · AnonReviewer2 · rating 4 · confidence 3 · 16 Dec 2016]
**Decent results, but unclear whether this is due to the proposed Stein variational gradient**

This paper considers the energy-based model interpretation of GAN, where the discriminator is an unnormalized model for the likelihood of a generative model p(x|theta) and the generator is a directed model that approximates this distribution. The generator is used to draw approximate negative phase samples that are used in stochastic maximum likelihood / contrastive divergence learning of the EBM / discriminator.

The main idea in the paper is to fit the generator by following the Stein variational gradient. In practice this gradient consists of the usual gradient provided by the discriminator with an added term that provides a repulsive force between the sampled data points to increase sample diversity.

The idea of using a kernel to push apart the sampled points is interesting, and will work in low dimensions, but it is hard to see how it can work in full scale images. For high dimensional samples x, the proposed kernel is unlikely to provide a useful distance measure between points. There are no convincing experiments in the paper that show otherwise. Specifically:

- There is no experiment that compares between standard GAN and GAN + repulsion, using the same architecture. (please address this in the rebuttal)
- If the Stein variational idea is taken literally, the right thing to do would be to fully optimize the generator at every step, and then taking a single optimization step on the discriminator. Instead, each is updated in turn, and the learning rates of both steps are adjusted to keep the two "in line".
- The kernel used to fit the generator is defined in the auto-encoder space of the discriminator, and thus depends on the discriminator parameters. The objective that is used to fit the generator thus changes at every step, and the procedure can no longer be interpreted as stochastic gradient descent with respect to any single well defined objective.

The authors obtain good results: The generated images clearly look better than those generated by DCGAN. However, their approach has a number of changes compared to DCGAN, so it is not clear where the improvement comes from. In addition, by now the DCGAN is no longer a very strong baseline, as various other techniques have been proposed.

Note: The use of phi for both the "particle gradient direction" and energy function is confusing

[Official Review · AnonReviewer1 · rating 4 · confidence 4 · 17 Dec 2016]
**Review: Learning to Draw Samples: With Application to Amortized MLE for Generative Adversarial Learning**

The authors propose amortized SVGD, an amortized form of prior work on SVGD, which is a particle variational method that maximally decreases the KL divergence at each update. "amortized SVGD" is done by training a neural network to learn this dynamic. They then apply this idea to train energy-based models, which admit a tractable unnormalized density.

In SVGD, the main difference from just MAP is the addition of a "repulsive force" that prevents degeneracy by encouraging probability mass to be spread to locations outside the mode. How this is able to still act as a strong enough entropy-like term in high dimensions is curious. From my understanding of their previous work, this was not a problem as the only experiments were on toy and UCI data sets.

In the experimental results here, they apply the kernel on the hidden representation of an autoencoder, which seems key, similar to Li et al. (2015) where their kernel approach for MMD would not work as well otherwise. However, unlike Li et al. (2015) the autoencoder is part of the model itself and not fixed. This breaks much of the authors' proposed motivation and criticisms of prior work, if they must autoencode onto some low-dimensional space (putting most effort then on the autoencoder, which changes per iteration) before then applying their method.

Unlike previous literature which uses inference networks, their amortized SVGD approach seems in fact slower than the non-amortized approach. This is because they must make the actual update on xi before then regressing to perform the update on eta (in previous approaches, this would be like having to perform local inferences before then updating inference network parameters, or at least partially performing the local inference). This seems quite costly during training.

I recommend the paper be rejected, and that the authors provide more comprehensive experimental results, expecially around the influence of the autoencoder, the incremental updates versus full updates, and the training time of amortized vs non-amortized approaches. The current results are promising but unclear why given the many knobs that the authors are playing with.

References

Li, Y., Swersky, K., & Zemel, R. (2015). Generative Moment Matching Networks. Presented at the International Conference on Machine Learning.

[Official Review · AnonReviewer3 · rating 4 · confidence 3 · 18 Dec 2016]
**Insufficient empirical evaluation.**

This paper proposes an amortized version of the Stein variational gradient descent (SVGD) method in which "a neural network is trained to mimic the SVGD dynamics". It applies the method to generative adversarial training to yield a training procedure where the discriminator is interpreted as an energy-based probabilistic model.

One criticism I have of the presentation is that a lot of time and energy is spent setting the table for a method which is claimed to be widely applicable, and the scope of the empirical evaluation is narrowed down to a single specific setting. In my view, either the paper falls short of its goal of showing how widely applicable the proposed method is, or it spends too much time setting the table for SteinGAN and not enough time evaluating it.

The consequence of this is that the empirical results are insufficient in justifying the approach proposed by the paper. As another reviewer pointed out, DCGAN is becoming outdated as a benchmark for comparison.

Qualitatively, SteinGAN samples don't look significantly better than DCGAN samples, except for the CelebA dataset. In that particular case, the DCGAN samples don't appear to be the ones presented in the original paper; where do they come from?

Quantitatively, DCGAN beats SteinGAN by a small margin for the ImageNet Inception Score and SteinGAN beats DCGAN by an even smaller margin for the CIFAR10 Inception Score. Also, in my opinion, the "testing accuracy" score is not a convincing evaluation metric: while it is true that it measures the amount of information captured in the simulated image sets, it is only sensitive to information useful for the discrimination task, not for the more general modeling task. For instance, this score is likely completely blind to information present in the background of the image.

Because of the reasons outlined above, I don't think the paper is ready for publication at ICLR.

[Author Response · Dilin Wang · 24 Jan 2017]
**Thank you for your review and comments**

We highly appreciate the time and feedback from all the reviewers, all of which we will take into serious consideration in our revision. We will particularly strengthen and clarify the empirical experiments. Below we address some of the major points: 

[Testing Accuracy Score]
We agree with the reviewers' point on the "testing accuracy" score, but think that it still provides some valuable insight about the dataset. Its blindness to the background can be a good thing in that it captures more information about the "effective amount" of objects the dataset contains.  The problem is that it is very difficult to obtain a *perfect* score, and reporting more than one metrics (in an objective fashion) can help to gain more comprehensive understandings. 

[Repulsive Term in High Dimension]
Our repulsive force works due to two tricks: 1) scaling the bandwidth with the data diversity using the median trick, which alleviates the exponential decay of RBF kernel. 2) define kernel on the feature space instead of the raw pixels of the images, which allows us to respect the manifold structure of the images. The framework of SVGD allows us to use any positive definite kernels and change it adaptively during iterations, because the kernel only defines the "tangent space" for improvement. 

SteinGAN without kernel corresponds to Viterbi training of the energy model and we find it work well with careful tuning of parameters, but tend to converge to a small number of bad-looking images after running a large number of iterations; adding the kernel under the same setting helps prevent this problem. Our current results on CIFAR10 shows that SteinGAN without kernel gives an inception score of 6.34, while that SteinGAN with kernel gives 6.76. 

[Amortized is slower than non-amortized]
Although the amortized algorithm has the overhead of updating $\xi$, it stores the information in a generative network, and allows us to simulate as many images as we need. By using the one-step gradient update we proposed, the update of $\xi$ is the same as standard backpropagation except replacing the Dlogp with the SVGD gradient.

[Final Decision · Program Chairs · 06 Feb 2017]
**ICLR committee final decision**

This paper presents an idea with a sensible core (augmenting amortized inference with per-instance optimization) but with an overcomplicated and ad-hoc execution. The reviewers provided clear guidance for how this paper could be improved, and thus I invite the authors to submit this paper to the workshop track.